# The Incidence and Risk Factors for Enterotoxigenic *E. coli* Diarrheal Disease in Children under Three Years Old in Lusaka, Zambia

**DOI:** 10.3390/microorganisms12040698

**Published:** 2024-03-29

**Authors:** Nsofwa Sukwa, Samuel Bosomprah, Paul Somwe, Monde Muyoyeta, Kapambwe Mwape, Kennedy Chibesa, Charlie Chaluma Luchen, Suwilanji Silwamba, Bavin Mulenga, Masiliso Munyinda, Seke Muzazu, Masuzyo Chirwa, Mwelwa Chibuye, Michelo Simuyandi, Roma Chilengi, Ann-Mari Svennerholm

**Affiliations:** 1Centre for Infectious Disease Research in Zambia (CIDRZ), Lusaka P.O. Box 34681, Zambia; nsofwa.sukwa@cidrz.org (N.S.); paul.somwe@cidrz.org (P.S.); monde.muyoyeta@cidrz.org (M.M.); kapambwe.mwape@cidrz.org (K.M.); kennedy.chibesa@cidrz.org (K.C.); chaluma.luchen@cidrz.org (C.C.L.); suwilanji.silwamba@cidrz.org (S.S.); bavin.mulenga@cidrz.org (B.M.); masichishimba@icloud.com (M.M.); seke.muzazu@cidrz.org (S.M.); masuzyo.chirwa@cidrz.org (M.C.); mwelwa.chibuye@cidrz.org (M.C.); michelo.simuyandi@cidrz.org (M.S.); chilengir@yahoo.com (R.C.); 2Department of Biostatistics, School of Public Health, University of Ghana, Accra P.O. Box LG13, Ghana; 3Department of Microbiology and Immunology, University of Gothenburg, 40530 Gothenburg, Sweden; ann-mari.svennerholm@microbio.gu.se

**Keywords:** ETEC, diarrhea, HIV, WASH

## Abstract

This study aimed to estimate the incidence and risk factors for Enterotoxigenic *Escherichia coli* (ETEC) diarrhea. This was a prospective cohort study of children recruited in a household census. Children were enrolled if they were 36 months or below. A total of 6828 children were followed up passively for 12 months to detect episodes of ETEC diarrhea. Diarrheal stool samples were tested for ETEC using colony polymerase chain reaction (cPCR). Among the 6828 eligible children enrolled, a total of 1110 presented with at least one episode of diarrhea. The overall incidence of ETEC diarrhea was estimated as 2.47 (95% confidence interval (CI): 2.10–2.92) episodes per 100 child years. Children who were HIV-positive (adjusted Hazard ratio (aHR) = 2.14, 95% CI: 1.14 to 3.99; *p* = 0.017) and those whose source of drinking water was public tap/borehole/well (aHR = 2.45, 95% CI: 1.48 to 4.06; *p* < 0.002) were at increased risk of ETEC diarrhea. This study found that children whose mothers have at least senior secondary school education (aHR = 0.49, 95% CI: 0.29 to 0.83; *p* = 0.008) were at decreased risk of ETEC diarrhea. Our study emphasizes the need for integrated public health strategies focusing on water supply improvement, healthcare for persons living with HIV, and maternal education.

## 1. Introduction

Enterotoxigenic *Escherichia coli* (ETEC) remains a significant public health concern, especially in low- and middle-income countries (LMICs). It is the third most common cause of moderate-to-severe diarrhea (MSD) in children under 5 years and travelers [1,2,3]. It is responsible for about 4.2% of diarrhea deaths in children under 5 years [4,5]. According to the World Health Organization (WHO), diarrheal diseases are the second leading cause of death in children under five, responsible for approximately 525,000 deaths annually [6]. In Zambia, the burden of ETEC diarrhea is especially high, having been one of the top five enteric pathogens detected among children under 5 years, with a prevalence of 40.7% [7]. Yet, detailed epidemiological data among children under three years is limited. This gap in knowledge hinders the development of effective interventions and policies aimed at reducing the incidence and severity of ETEC diarrhea.

Understanding the epidemiology of ETEC diarrhea in young children is crucial for the identification of specific risk factors associated with the disease, which can inform targeted prevention and control strategies. Considering the evolving nature of bacterial pathogens and changing environmental and socio-economic conditions, contemporary data are essential to track the current state of the disease and its determinants. Studies have shown that interventions focusing on improving water quality, sanitation, and hygiene (WASH), alongside vaccination and healthcare access improvements, can significantly reduce the incidence of diarrheal diseases [8]. The Zambian context provides a unique setting to study ETEC diarrhea due to its specific demographic, environmental, and socio-economic characteristics, which may offer insights applicable to other similar regions.

ETEC infection typically occurs through the fecal–oral route, often stemming from the ingestion of contaminated food or water. Upon entry into the body, the bacterium targets the small intestinal epithelium, where it adheres and colonizes using specific colonization factors. Subsequently, ETEC releases enterotoxins, namely heat-labile toxin (LT) and heat-stable toxin (ST), both of which contribute to the manifestation of symptoms [9]. It is noteworthy that ETEC strains can express either one or both toxins, and the combination of enterotoxins and colonization factors can vary significantly based on geographical location. Therefore, it is crucial to comprehend the prevalent strains in a given region [10].

These enterotoxins and colonization factors serve as the primary virulence determinants of ETEC, playing pivotal roles in its pathogenicity. Moreover, their immunogenic properties make them attractive targets for vaccine development [11]. Understanding the diversity and distribution of these virulence factors is essential for designing effective preventive measures against ETEC-associated diarrheal diseases.

This study aims to fill the knowledge gap by providing up-to-date, specific data on the burden of ETEC diarrhea in this vulnerable age group. The specific objectives are to estimate the incidence of ETEC diarrhea and identify its associated risk factors among children under 36 months and below in Zambia. Through a prospective cohort design, we sought to understand the frequency of the disease and the influence of various risk factors, such as HIV status, water source quality, and maternal education level. The findings of this study are expected to inform public health policies and intervention strategies, ultimately contributing to the reduction in ETEC diarrhea’s impact on child health in Zambia and similar settings.

## 2. Materials and Methods

### 2.1. Study Design and Participants

This was a prospective cohort study design in which children 36 months and below were recruited in a household census conducted within the catchment area of five health facilities in the Lusaka urban district, namely: Matero General Hospital, Chawama General Hospital, George Clinic, Kanyama General Hospital, and Chainda South clinic. These health facilities cater to populations that can be considered representative of Lusaka’s peri-urban communities. To meet the sample size, the catchment area for the health facility was defined to be within a 2.0 km radius of the facility. Study participants were residents in the catchment areas of the health facility for at least 6 months. A surveillance system was set up at each facility’s outpatient department (OPD) for passive case detection of diarrhea among children ≤ 36 months of age registered during the household census. To detect any seasonal variations of disease burden, the surveillance system was conducted for a total of 12 months, and the last child to be enrolled was followed up for 9 months.

### 2.2. Study Procedures

During the household census, informed consent for participation in the household census was obtained from the Head of the Household or a designee via an in-person interview data, including sociodemographic characteristics, water, sanitation, and hygiene (WASH), nutrition, and health-seeking behavior. Children aged 36 months and below were enrolled in the cohort study and provided with study-specific identity (ID) cards. These children constituted the population at risk of ETEC diarrhea and the denominator for the incidence of ETEC diarrhea.

At the health facility, any child presenting with diarrhea, aged 36 months and below, in possession of a study ID card and accompanied by a legally authorized representative willing to provide written informed consent and a stool sample for testing was included in the passive surveillance. A child was excluded if they were not in possession of a valid study ID card and if they were born after the completion of the census stage of the study. A second consent was provided at the clinic, and information about the participant was collected using the diarrhea surveillance case report form (CRF). This included medical history, clinical presentation, and physical examination, followed by routine management of diarrhea. Participants were treated with zinc supplementation and fluid replacement in the form of oral rehydration salts or intravenous fluids where appropriate. Antibiotics were also given when indicated, i.e., in the case of bloody diarrhea, for treatment of concurrent illness, e.g., respiratory tract infection. A stool sample was collected from each participant presenting with diarrhea at the health facility prior to treatment or after, depending on how ill the child was during the clinic visit. Diarrhea was defined as having ≥3 episodes of looser-than-normal stools in 24 h.

The household census data were collected on Android tablets using Open Data Kit (ODK^®^ Collect v1.27 Beta 2020), which facilitated the collection and submission of data to an ODK^®^ aggregate server. Validation checks were built into the electronic ODK^®^ data collection form to ensure real-time checks for valid values, appropriate skip patterns, and logic consistency. Databases storing the various study data were password-protected and backed up daily. The research assistants collecting the data were trained in human research ethics, data collection, and administering the questionnaire.

Surveillance data were collected and then entered into a study-specific CRF programmed into the District Health Information System 2 (DHIS2) database. The study ID, which was assigned to each eligible child during the census, was captured for every participating child in the database. This ensured that the surveillance data for each child could be uniquely linked to their household census data. The quality of data collected was checked daily and identified errors were corrected immediately. All data collected were anonymized and ID numbers were the only means of identifying the study participants. All data collected were stored on a password-protected central server. Both the database and data collection forms were stored securely and access to them was restricted to authorized study staff only.

This study was reviewed and approved by the University of Zambia Biomedical Ethics Review Committee (UNZABREC Ref: 1091-2020) and the National Health Research Authority (NHRA).

### 2.3. Laboratory Procedures

Fresh stool samples were collected in sterile containers and transported to the laboratory at 2–8 °C within 8 h. Enterotoxigenic *E. coli* was identified using colony polymerase chain reaction (PCR) preceded by standard microbiological identification techniques including culture and biochemical tests and described previously [12]. Upon reception, the stool was streaked for isolation on MacConkey agar (Oxoid, Hampshire, UK) using a sterile inoculation loop. The plates were then incubated for 24 h at 37 °C. The following day, five distinct lactose fermenting colonies were sub-cultured onto new MacConkey (Oxoid, Hampshire, UK) culture plates to obtain pure colonies. Twenty-four hours after incubation, the colonies were subjected to the following biochemical tests: triple sugar iron (TSI) (TM Media, Delhi, India), lysine iron agar (LIA) (TM Media, Delhi, India), and sulfide indole motility (SIM) (TM Media, Delhi, India) tests and incubated aerobically at 37 °C for 24 h. All colonies with a characteristic *E. coli* reaction were subjected to colony PCR for toxins and phenotypic dot blot assay for colonization factor identification. Nucleic acid extraction, primers (Appendix A), and PCR conditions were conducted using the method described previously [13]. Briefly, DNA extraction was conducted using the boiling method. A loopful of bacteria was suspended in molecular grade nuclease-free water (Invitrogen, Waltham, MA, USA) and heated to 100 °C for 10 min and span to extract the supernatant. The resulting DNA served as a template for PCR amplification to detect ETEC using a multiplex PCR assay targeting the heat-labile toxin (LT) and heat stable toxin (STp and STh) genes. Molecular-grade water (Invitrogen, MA, USA) was used as a negative control in all PCR reactions alongside internal ETEC positive control. The PCR reaction mixture contained template DNA, specific primers, MgCl (Invitrogen, MA, USA) and ReadyMix™. The PCR was conducted with specific thermal cycling conditions. Amplicons were analyzed by electrophoresis on agarose gel (Fischer Scientific, Geel, Belgium) and visualized using UV light after staining. To identify the CFs, a monoclonal antibody dot blot assay [13] was used. Stored ETEC isolates at −80 °C were revived by thawing at room temperature and inoculating them on MacConkey agar (Oxoid, Hampshire, UK) plates. After a 24 h incubation, the isolates were sub-cultured onto colonization factor antigen (CFA) agar plates and incubated at 37 °C for 24 h. The dot blot assay was performed as described previously [13]. Briefly, 2 μL of bacterial suspensions at a density of approximately 2 × 10^9^ bacteria per ml of phosphate-buffered saline (PBS) (Sigma Aldrich, St. Louis, MO, USA) was applied as a dot to a nitrocellulose membrane. The membrane was allowed to dry for 5 min and blocked with bovine serum albumin-PBS for 20 min. Monoclonal antibody (MAb) against CF diluted in 0.1% bovine serum albumin-PBS–0.05 Tween 20 was then added (Sigma Aldrich, MO, USA), and the membrane was incubated overnight in a humid chamber and washed with PBS–0.05% Tween. (Sigma Aldrich, MO, USA) Thereafter, a goat anti-mouse immunoglobulin G horseradish peroxidase conjugate solution was added to the membrane and incubated for 2 h in a humid chamber; the bound MAb was detected by the addition of 4-chloro-1-naphthol chromogen (Sigma Aldrich, MO, USA) in Tris-buffered saline (Sigma Aldrich, MO, USA) and H_2_O_2_ (Sigma Aldrich, MO, USA). All incubations were performed at room temperature. A dark blue or grey dot on the strip was interpreted as positive. The CFs tested for in our assay were CFA/I, CS1, CS2, CS3, CS4, CS5, CS6, CS7, CS12, CS14, and CS17 using a separate nitrocellulose strip for detection of each CF. The MAbs used were developed and provided by Gothenburg University, Gothenburg, Sweden.

### 2.4. Definitions

The head of the household was defined as the person all members of the household regard as the head. He/she is responsible for making the day-to-day decisions governing the running of the household.

Access to improved water was defined according to the household’s use of the following types of water supply for drinking: piped water, public tap, borehole or pump, and protected well. Improved water sources did not include unprotected wells and springs, rivers, or ponds.

Bottled water was also excluded as it could not be determined if the water was bottled under safe conditions.

Water was considered as treated if it was boiled or chlorinated. All other methods were also captured but were not considered as ‘treatment’. These other methods included leaving water in the sun to disinfect and filter through cloths, ceramic, or another filter.

Stored water was any water that was placed in a container (clay pot, cooking pot, jerrycan, plastic bottle, or drum) for any duration after it was collected from the source.

Improved sanitation facilities for the household included drinking water, which was piped into the house or yard, connection to a public sewer or septic system, flush toilet, and handwashing behavior at key time points. Unimproved sanitation facilities included drinking water from public taps/boreholes or wells, pit latrines, or no facilities.

Hand washing was assessed by open-ended questions where participants would mention important time points for handwashing. Participants were also directly observed to record whether soap was available and whether they used it or not.

ETEC diarrhea was defined as ≥3 looser than normal stools in 24 h with a positive cPCR result.

### 2.5. Statistical Analysis

The sample size was calculated based on estimating ETEC incidence with a certain level of precision. A total of 30 events (new cases of confirmed ETEC), corresponding to an expected sample size of 1000, produces a two-sided 95% confidence interval with a width equal to 0.025 when the estimate of λ (the hazard rate) is 0.035. The percentage of censoring is anticipated to be 97%. The estimates assumed type-II censoring, in which the participants are followed up until 30 failures occur. For a hazard rate of 0.04, 40 events were required, corresponding to an expected sample size of 1334 participants under similar assumptions. It was assumed that all sites had similar incidences. Therefore, each of the 5 sites enrolled at least 1350 children, bringing the total sample size for the study to 6750 participants.

Background characteristics were summarized using frequency and percentage for categorical variables, while median and interquartile interval (IQI) were used for continuous variables. WASH was defined according to the methodology of the Zambia Demographic Health Survey (ZDHS), while anthropometric indices (i.e., stunting, wasting, and underweight) indices were defined according to the methodology of the World Health Organization using gender, age, weight, mid-upper arm circumference (MUAC), and height or length data collected during the census.

The incidence of ETEC diarrhea was estimated as the total number of ETEC diarrhea episodes divided by total child-year at the end of one year of follow-up. The Andersen–Gill (AG) model (i.e., Cox with robust standard error) was used to identify factors independently associated with the risk of ETEC diarrhea. This model adjusted the standard error to account for the clustering of episodes within a child. To implement the AG model, the dataset was set up such that, for each patient, there was one observation per event or time interval. For example, if a child has one event, then there were two observations for that child. The first observation covered the time span from the date of enrolment into the study until the time of the event, and the second observation covered the time from the event to the end of follow-up. The outcome was defined as “1” if the child had ETEC diarrhea and “0” if otherwise. Children who did not present with diarrhea at the clinical research facility during the study follow-up period were considered as not having ETEC diarrhea and were censored at the end of the follow-up. In building a parsimonious model for risk factors for ETEC diarrhea, all variables that were univariably associated with the incidence of ETEC diarrhea at *p* ≤ 0.1 were considered for inclusion in the multivariable AG model. Variables were removed at *p* ≤ 0.1. The final model was verified using a backward stepwise selection algorithm. All analyses were performed using Stata 17 SE (Stata Corp, College Station, TX, USA).

To assess the severity of diarrhea, three scoring methods were utilized, namely DHAKA [14], VERSIKARI [15], and the CIDRZ score [16], which is a composite score based on the parameters included in the DHAKA, CLARK, WHO, and VERSIKARI.

## 3. Results

### 3.1. Sociodemographic and Household Characteristics of Participants

A total of 6602 households were surveyed. Among these, 6828 children were eligible and enrolled across all study sites. The total number of participants enrolled was equally distributed among the sites, with each site having just over 1350 (20%) participants. Overall, 49.4% (*n* = 3374) were female and 50.6% (*n* = 3454) were male; 15.1% (*n* = 1032) were under 6 months; 19.8% (*n* = 1349) were aged between 6 and <12 months; 19.0% (*n* = 1299) were aged between 12 and <18 months; 17.1% (*n* = 1167) were aged between 18 and <24 months; and 29.0% (*n* = 1981) were aged between 24 and 30 months. The majority of children surveyed during the census were stunted at 55.1% (*n* = 3762). However, only 4.2% (*n* = 286) and 14.5% (*n* = 991) were wasted and underweight, respectively. Overall, 81.7% *n* = 5581 had a normal MUAC and 5.1% (*n* = 347) had a MUAC less than 12.5 cm, making them severely malnourished (Table 1).

The median size of each household in each of the five catchment areas was five members (Interquartile Interval (IQI): 4–5). The majority of the households were two-parent (82.0%), male-headed (male 79.9%, female 20.1%) with a median age of 35 years (IQI: 30–43 years), and married (*n* = 5656, 85.7%). The primary caregiver for the majority of the participants was the mother 6007 (88.0%), followed by a close female relative (Table 1).

The main source of drinking water for most households was a public tap or public borehole 53.1% (*n* = 3629), water piped into the yard 44.7% (*n* = 3052), and the main toilet facility was a pit latrine. Overall, the WASH facilities in all five communities were unimproved.

### 3.2. Incidence of Enterotoxigenic Escherichia coli (ETEC) Diarrhea and Associated Risk Factors

Among the 6828 eligible children enrolled, 1582 visited the facility, while 5249 children did not visit the health facility and were not followed up after the completion of the census (Figure 1). Of the 1582 who visited the facility, a total of 1110 presented with at least one episode of diarrhea. Of these, 121 children had at least one episode of ETEC diarrhea (Figure 1). A total of 141 episodes of ETEC diarrhea were observed and the total child-years of follow-up was 5697.37 (Table 2).

The overall incidence of ETEC diarrhea was estimated at 2.47 episodes per 100 child years (Table 2). Children who were HIV-positive and those whose source of drinking water was public tap/borehole/well were at increased risk of ETEC diarrhea (aHR = 2.14, (95% CI): 1.14 to 3.99; *p* = 0.017) and (aHR = 2.45, 95% CI: 1.48 to 4.06; *p* < 0.002) respectively (Table 2). It was also observed that children whose mothers have at least senior secondary school education (aHR = 0.49, 95% CI: 0.29 to 0.83; *p* = 0.008) had a decreased risk of ETEC diarrhea (Table 2).

### 3.3. Enterotoxigenic Escherichia coli (ETEC) Characterization

ETEC positive for the heat-labile toxin (LT) only (47.2%, *n* = 59) was the most prevalent among the ETEC isolates tested in this study, followed by ST only (36%, *n* = 46) and LT/ST (16%, *n* = 20). Outputs from the PCR were visualized using 1.5% agarose gel, see Appendix A. The most frequently detected CFs were CS2/CS3 (9.6%, *n* = 12) followed by CS6 (8.0%, *n* = 10) and CS14 (5.6%, *n* = 7). The results from the ETEC characterization are summarized in Table 3.

### 3.4. Enterotoxigenic Escherichia coli (ETEC) Diarrhea Disease Severity

Severity was assessed using the four scoring methods that were previously mentioned, and most of the samples were from children with mild diarrhea. Based on the DHAKA scoring, 2.84% of the 141 ETEC-positive diarrhea episodes were from children with severe diarrhea, while 3.55% and 91.49% of the samples were from children with moderate and mild diarrhea, respectively. The proportion of children with either moderate or severe ETEC-associated diarrhea was similar (about 7%) across severity scores. The results are shown in Figure 2 below.

## 4. Discussion

This was a prospective cohort study of children recruited in a household census to estimate the incidence and associated risk factors for ETEC diarrhea in Zambian children aged 36 months or below. The overall incidence of ETEC diarrhea was estimated at 2.47 episodes per 100 child years. It was found that children who were HIV-positive were about three times more likely to have ETEC diarrhea compared to their HIV-negative counterparts, while those whose source of drinking water was a public tap or borehole or well had over twice the risk of ETEC diarrhea compared to those with piped water into the house or yard. Furthermore, children whose mothers have at least senior secondary school education had about 51% decreased risk of ETEC diarrhea compared to those whose mothers have less than secondary education.

Our estimated incidence of ETEC diarrhea is supported by previous studies that identified ETEC as a major cause of moderate to severe diarrhea among children in low- and middle-income countries. For instance, a study by Chisenga et al. (2018) in Zambia reported ETEC as the third most detected pathogen among children under five [7]. This underscores the global burden of ETEC in low-resource settings, driven largely by factors like poor sanitation and limited access to clean water. While our study detected a lower-than-expected number of moderate-to-severe diarrhea cases, it could be attributed to a few factors related to the way the study was conducted. During the census, mothers were encouraged to report to the health facility as soon as their child had diarrhea, and once at the facility, dedicated staff attended to them, leading to less time spent at the facility. These motivations could have led mothers to bring their children to the facilities early before the disease progressed to moderate or severe disease.

ETEC causes disease by adhering to intestinal cells using colonization factors (CF) and releasing toxins, heat-stable (ST) and/or heat-labile enterotoxin (LT) [11,17]. Among over 25 CFs [18], a few, like CFA/I, CS1, CS2, CS4, CS3, CS5, CS6, CS7, CS14, CS17, and CS21, are the most common [19,20,21]. In our study, half of the ETEC strains expressed only LT toxin, a third only ST toxin, and the rest both LT and ST. The most frequent CFs were CS2 + CS3, often associated with LT/ST toxins, followed by CS6 linked to ST. Toxin distribution has been shown to vary by geographical location with some strains being more common than others across regions. These results on toxin distribution are consistent with previous studies conducted in Zambian children [22] and in Jamaica [23], but they found a higher occurrence of LT-only strains than in some other studies [1,24,25,26,27].

The increased risk in HIV-positive children is consistent with previous studies. In a study conducted early in the HIV epidemic, the incidence of diarrhea was higher in HIV-infected infants when compared to HIV-uninfected infants [28,29]. Furthermore, HIV-infected infants are more likely to have recurrent, persistent, and more severe diarrhea than HIV-uninfected infants [30,31,32]. When combined with what is already documented on diarrhea sequelae such as malnutrition, these findings further reinforce the vulnerability of immunocompromised children to enteric infections. This association highlighted the compounded vulnerability of HIV-positive children to enteric pathogens, necessitating targeted health interventions.

The association between unimproved water sources and increased ETEC diarrhea risk supports global health observations, emphasizing the role of water quality in diarrheal diseases. The Global Burden of Disease report highlights unsafe water as a leading risk factor for diarrheal diseases worldwide [33]. Poor WASH significantly contributes to increased diarrhea mortality and morbidity, malnutrition, environmental enteropathy, and iron deficiency anemia [34]. In Zambia, 11.4% of deaths are due to WASH-associated factors [35]. Although recent reports indicate improvements in WASH, with 72.3% and 54.4% of the population having access to improved water sources and sanitation, respectively, diarrheal disease outbreaks still persist in peri-urban areas, suggesting that there exists poor WASH in unplanned areas [36]. Our study contributes to this body of evidence, emphasizing the critical need to improve water quality in developing countries to mitigate the burden of waterborne diseases like ETEC diarrhea.

The positive effect of maternal education with regard to decreased incidence of ETEC diarrhea suggests the critical role of maternal health literacy in disease prevention. This aligns with broader public health research, indicating that maternal education positively impacts child health outcomes. Previous studies have shown a positive association between a mother’s educational attainment and a child’s health indicators, such as weight-for-age, stunting, and wasting, as well as the child’s health behavior [37,38]. Maternal education has also been positively associated with childhood immunization, iron supplementation during pregnancy, and uptake of medical services [39,40]. This link suggests that maternal education might indirectly influence health behaviors and access to healthcare, thereby reducing the risk of diarrheal diseases in children.

Our study has several strengths, including a large sample size drawn from household censuses in the entire health facility catchment area population and the one-year-long follow-up. This enhances the reliability of our findings. The use of cPCR for ETEC detection represents a methodological strength in relation to previously used phenotypic analysis accuracy. However, our study also has some limitations, including the passive surveillance in detecting ETEC diarrhea, which potentially could have underestimated ETEC diarrhea incidence due to unreported cases. Additionally, the exclusion of children who did not visit the health facility may have introduced selection bias.

Our findings have critical implications for public health interventions in Zambia. The identification of risk factors, particularly HIV status and water source, suggests targeted intervention areas. Improving water quality and accessibility, alongside focused healthcare for HIV-positive children, could substantially reduce ETEC incidence. The role of maternal education highlights the need for educational programs, potentially offering a long-term strategy to reduce diarrheal diseases. Current vaccine formulations in development target the LT toxin and the most prevalent CFs. Results from the GEMS study have shown that vaccine candidates covering strains expressing the major CFs (i.e., CFA/I and CS1-6) can be highly efficacious and prevent up to 66% of ST only and LT/ST strains even with some geographic variation [41]. Furthermore, there are many shared cross-reactive epitopes for which a vaccine could be formulated to cover 4 to 5 of the most common colonization factors associated with illness in infants and travelers [11].

## 5. Conclusions

Our study contributes significantly to understanding the epidemiology of ETEC diarrhea in Zambian children under three years. It underscores the need for integrated public health strategies focusing on water quality improvement, healthcare for vulnerable groups, and maternal education. Such multifaceted approaches are essential for reducing the burden of ETEC diarrhea and enhancing child health outcomes in low- and middle-income countries.

## Figures and Tables

**Figure 1 microorganisms-12-00698-f001:**
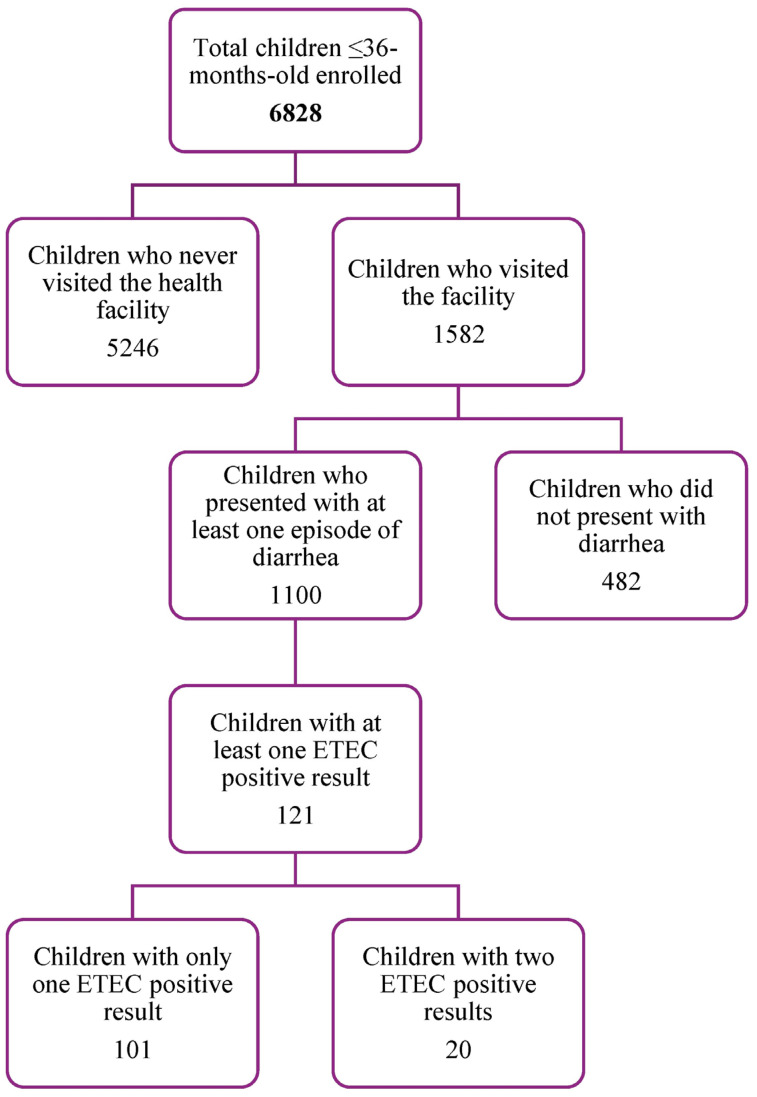
Summary of the surveillance study flow.

**Figure 2 microorganisms-12-00698-f002:**
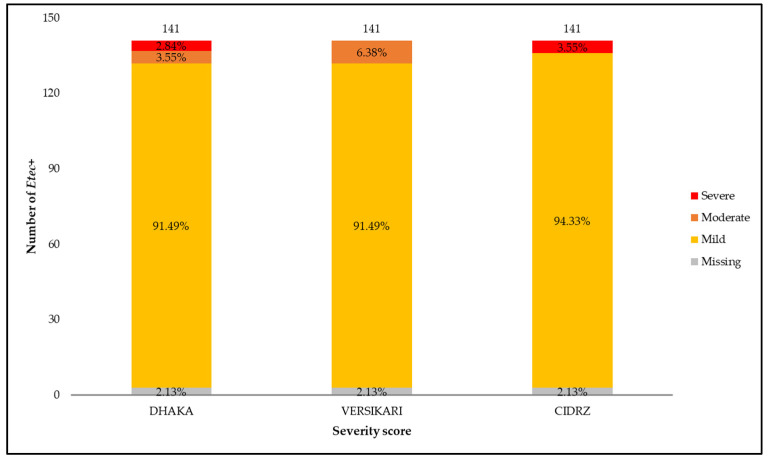
Diarrheal disease severity for all ETEC diarrhea cases.

**Table 1 microorganisms-12-00698-t001:** Characteristics of study participants.

	Children Enrolled N (% of Total)
Characteristics	6828 (100)
Catchment area	
Chainda-South	1371 (20.1)
Chawama	1370 (20.1)
George	1364 (20.0)
Kanyama	1358 (19.9)
Matero	1365 (20.0)
Household size	
<5	2935 (43.0)
5+	3893 (57.0)
Age group in months	
0 to 5	1032 (15.1)
6 to <12	1349 (19.8)
12 to <18	1299 (19.0)
18 to <24	1167 (17.1)
24 to 36	1981 (29.0)
Gender	
Male	3454 (50.6)
Female	3374 (49.4)
Stunted *	
No	2220 (32.5)
Yes	3762 (55.1)
Missing	846 (12.4)
Wasted *	
No	5504 (80.6)
Yes	286 (4.2)
Missing	1038 (15.2)
Underweight *	
No	5021 (73.5)
Yes	991 (14.5)
Missing	816 (12.0)
MUAC *	
Normal: >12.5 cm	5581 (81.7)
Moderate and severe acute malnutrition: ≤12.5 cm	347 (5.1)
Missing	900 (13.2)
Child HIV status	
Negative	6107 (89.4)
Positive	370 (5.4)
Missing	351 (5.1)
Rotarix vaccination status *	
No—1 dose only or not vaccinated	5536 (81.1)
Yes—Received both doses	1169 (17.1)
Missing	123 (1.8)
Mother breastfeed child	
Yes	6660 (97.5)
No	120 (1.8)
Missing	48 (0.7)
Mothers HIV status	
Negative	5582 (81.8)
Positive	1003 (14.7)
Missing	243 (3.6)
Primary caregiver *	
Mother	6006 (88.0)
Other	822 (12.0)
Mother’s highest level of education *	
None/Primary/Junior Secondary/Other	4194 (61.4)
Senior Secondary/University/Tertiary	2634 (38.6)
Drinking water source *	
Piped into house/Piped water into yard	3052 (44.7)
Public tap/Borehole/Well	3629 (53.1)
Sachet/Bottled/Filtered/Other	147 (2.2)
Treated water	
No	3915 (57.3)
Yes	2913 (42.7)
Toilet facility	
Pit latrine	4548 (66.6)
Flush toilet	2211 (32.4)
No facility	69 (1.0)
WASH *	
Improved toilet and water and good hand washing	1657 (24.3)
Unimproved toilet and water and bad hand washing	5171 (75.7)
Head of Household gender	
Male	5430 (79.5)
Female	1398 (20.5)
Household head marital status *	
Married/Cohabiting	5820 (85.2)
Single/Divorced/Separated/Widowed	1008 (14.8)
Household head age group	
17–35	3481 (51.0)
36–49	2445 (35.8)
50+	902 (13.2)

* denotes newly generated variables and variables whose categories were re-categorized. MUAC—mid-upper arm circumference, WASH—Water, sanitation, and hygiene.

**Table 2 microorganisms-12-00698-t002:** Risk factors for ETEC diarrhea.

	ETEC + Diarrhea Episodes *n*	Number of Child Years Y	Incidence Rate per 100 CY IR (95% CI)	Univariable HR (95% CI)	*p*-Value	Multivariable aHR (95% CI)	*p*-Value
Characteristics	141	5697.37	2.47 (2.10–2.92)	-	-	-	-
Household size							
<5	70	2435.4	2.87 (2.27–3.63)	1	0.145		
5+	71	3262	2.18 (1.72–2.75)	0.76 (0.52–1.1)			
Age group in months							
0 to 5	28	865.85	3.23 (2.23–4.68)	1	0.083		
6 to <12	32	1125.83	2.84 (2.01–4.02)	0.88 (0.49–1.57)			
12 to <18	33	1073.43	3.07 (2.19–4.32)	0.95 (0.54–1.66)			
18 to <24	23	968.68	2.37 (1.58–3.57)	0.73 (0.4–1.36)			
24 to 36	25	1663.58	1.5 (1.02–2.22)	0.46 (0.26–0.85)			
Gender							
Male	67	2882.1	2.32 (1.83–2.95)	1	0.515		
Female	74	2815.3	2.63 (2.09–3.3)	1.13 (0.78–1.64)			
Stunted *							
No	41	1858.84	2.21 (1.62–3)	1	0.356		
Yes	85	3156.85	2.69 (2.18–3.33)	1.22 (0.8–1.87)			
Wasted *							
No	117	4604.86	2.54 (2.12–3.05)	1	0.918		
Yes	6	246.92	2.43 (1.09–5.41)	0.96 (0.42–2.17)			
Underweight *							
No	96	4198.07	2.29 (1.87–2.79)	1	0.027		
Yes	31	840.41	3.69 (2.59–5.25)	1.62 (1.05–2.47)			
MUAC *							
Normal: >12.5 cm	115	4682.47	2.46 (2.05–2.95)	1	0.072		
Moderate and severe acute malnutrition: ≤12.5 cm	13	293.47	4.43 (2.57–7.63)	1.81 (0.95–3.45)			
Child HIV status							
Negative	115	5092.48	2.26 (1.88–2.71)	1	<0.001	1	0.017
Positive	20	318.55	6.28 (4.05–9.73)	2.78 (1.64–4.72)		2.14 (1.14–3.99)	
Rotarix vaccination status at 6 and 10 weeks *							
No—1 dose only or not vaccinated	112	4629.62	2.42 (2.01–2.91)	1	0.364		
Yes—Received both doses	29	972.38	2.98 (2.07–4.29)	1.23 (0.79–1.93)			
Mother breastfeed child							
Yes	140	5557.64	2.52 (2.13–2.97)	1	0.351		
No	1	101.23	0.99 (0.14–7.01)	0.39 (0.06–2.8)			
Mothers HIV status							
Negative	102	4664.76	2.19 (1.8–2.65)	1	<0.001		
Positive	37	836.3	4.42 (3.21–6.11)	2.03 (1.34–3.06)			
Primary caregiver *							
Mother	132	5010.11	2.63 (2.22–3.12)	1	0.065		
Other	9	687.26	1.31 (0.68–2.52)	0.5 (0.24–1.04)			
Mother’s highest level of education *							
None—Junior Secondary	111	3511.2	3.16 (2.62–3.81)	1	<0.001	1	0.008
Senior Secondary—Tertiary	30	2186.2	1.37 (0.96–1.96)	0.44 (0.28–0.68)		0.49 (0.29–0.83)	
Drinking water source *							
Piped water—house/yard	28	2516.62	1.11 (0.77–1.61)	1	<0.001	1	
Public tap/Borehole/Well	107	3053.92	3.5 (2.9–4.23)	3.15 (1.99–4.99)		2.45 (1.48–4.06)	0.002
Sachet/Bottled/Filtered/Other	6	126.83	4.73 (2.13–10.53)	4.3 (1.59–11.65)		2.99 (0.84–10.57)	
Anything done to water to make it safe to drink							
No	78	3250.6	2.4 (1.92–3)	1	0.699		
Yes	63	2446.7	2.57 (2.01–3.3)	1.08 (0.74–1.56)			
Toilet facility							
Pit latrine	116	3820.35	3.04 (2.53–3.64)	1	0.002		
Flush toilet	22	1819.85	1.21 (0.8–1.84)	0.4 (0.23–0.68)			
No facility	3	57.18	5.25 (1.69–16.27)	1.72 (0.56–5.28)			
WASH *							
Improved toilet and water and good hand washing	15	1358.1	1.1 (0.67–1.83)	1	0.003		
Unimproved toilet and water and bad hand washing	126	4339.3	2.9 (2.44–3.46)	2.63 (1.4–4.93)			
Head of Household gender							
Male	106	4527.1	2.34 (1.94–2.83)	1	0.265		
Female	35	1170.3	2.99 (2.15–4.17)	1.28 (0.83–1.98)			
Household head marital status *							
Married/Cohabitating	123	4853.27	2.53 (2.12–3.02)	1	0.539		
Single/Divorced/Separated/Widowed	18	844.1	2.13 (1.34–3.38)	0.84 (0.49–1.45)			
Household head age group							
17–35	81	2901.55	2.79 (2.25–3.47)	1	0.391		
36–49	44	2034.54	2.16 (1.61–2.91)	0.78 (0.51–1.17)			
50+	16	761.28	2.1 (1.29–3.43)	0.76 (0.42–1.37)			

* denotes newly generated variables and variables whose categories were re-categorized. Univariable effect estimates were computed on non-missing observations. Multivariable effect estimates were computed on variables that yielded a *p*-value ≤ 0.1 at univariable analysis. Only adjusted statistically significant effect (*p* ≤ 0.05) estimates have been presented. N = 5555.

**Table 3 microorganisms-12-00698-t003:** Number and frequencies (%) of ETEC isolates with respective enterotoxin and colonization factor profiles.

Enterotoxin	Total *n *= 125 *	None (*n* = 80, 64%)	CFA/I (*n* = 1, 0.8%)	CS1, CS3 (*n* = 1, 0.8%)	CS2, CS3 (*n* = 12, 9.6%)	CS3 (*n* = 2, 1.6%)	CS5, CS6, CS7 (*n* = 4, 3.2%)	CS17 (*n* = 3, 2.4%)	CS6 (*n* = 10, 8%)	CS7 (*n* = 4, 3.2%)	CS14 (*n* = 7, 5.6%)	CS17 (*n* = 1, 0.8%)
	*n* (%)	*n* (%)	*n* (%)	*n* (%)	*n* (%)	*n* (%)	*n* (%)	*n* (%)	*n* (%)	*n* (%)	n (%)	*n* (%)
LT	59 (47.2)	44 (74.6)	0 (0.0)	0 (0.0)	3 (5.1)	0 (0.0)	1 (1.7)	3 (5.1)	4 (6.8)	3 (5.1)	1 (1.7)	0 (0.0)
LT/STh	13 (10.4)	5 (38.5)	0 (0.0)	0 (0.0)	6 (46.2)	2 (15.4)	0 (0.0)	0 (0.0)	0 (0.0)	0 (0.0)	0 (0.0)	0 (0.0)
LT/STp	7 (5.6)	4 (57.1)	0 (0.0)	0 (0.0)	2 (28.6)	0 (0.0)	0 (0.0)	0 (0.0)	1 (14.3)	0 (0.0)	0 (0.0)	0 (0.0)
STh	25 (20.0)	13 (52.0)	0 (0.0)	1 (4.0)	1 (4.0)	0 (0.0)	2 (8.0)	0 (0.0)	2 (8.0)	0 (0.0)	6 (24.0)	0 (0.0)
STh/STp	1 (0.8)	1 (100.0)	0 (0.0)	0 (0.0)	0 (0.0)	0 (0.0)	0 (0.0)	0 (0.0)	0 (0.0)	0 (0.0)	0 (0.0)	0 (0.0)
STp	20 (16.0)	13 (65.0)	1 (5.0)	0 (0.0)	0 (0.0)	0 (0.0)	1 (5.0)	0 (0.0)	3 (15.0)	1 (5.0)	0 (0.0)	1 (5.0)

* Total number of isolates stored in 15% glycerol that were analyzed for toxins and CFs was reduced from 141 to 125 as some isolates could not be revived during testing.

## Data Availability

Requests for access to the data supporting the reported findings can be made by contacting the corresponding author.

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
