# Peer review of "The Incidence and Risk Factors for Enterotoxigenic E. coli Diarrheal Disease in Children under Three Years Old in Lusaka, Zambia"

_microorganisms, 2024, doi:10.3390/microorganisms12040698_

Round 1

Reviewer 1 Report

Comments and Suggestions for Authors

Dear Editor and authors,

Despite no new novelty regarding hypothesis and results obtained (a certain obviousness in the answers), the manuscript was very well written and presented. The entire process was explained following the "to-do" list in a good manuscript.

I have some minor suggestions before recommend the publication, as follows:

L27: adjusted with non-capital initial.

References in text: brackets not parenthesis.

L42: 525,000.

L74: space between numbers and units.

L132: space between numbers and units and so on in all text.

L138: Sigma Aldrich.

L165: Paragraph.

Figure 1: ETEC in uppercase.

Table 1: gender, not sex. And in all text and tables.

Table: University tertiary or University/tertiary.

L236: 1582.

L238: 1582, 1110 and so on.

Table 2: Why 0.145 and 0.083 in bold? Are they significant at statistical level?

L333: Trogler et al.? MDPI Guidelines.

L409: Please formatt.

L453: Please re-formatt.

Congratulations.

Reviewer 2 Report

Comments and Suggestions for Authors

Dear editor

Concerning the MS The incidence and risk factors for Enterotoxigenic E. coli diarrhoeal disease in children under 3 years old in Lusaka, Zambia

It includes the epidemiological data of Enterotoxigenic E. coli diarrhoeal disease in children under 3 years old in Lusaka, Zambia with some genetic data and serological studies of ETEC

Some points need to be elucidated

Introductions

More details are need to be added to the introduction concerning the toxicity of the EXEC, the types of toxins the virulence factors

Methods

-        Please indicate the inclusion and the exclusion criteria

-        How the colony PCR was performed

-        Mention the method, primers and used Kits

-        Is the children take any medication the study did not mention how the diarrhea was controlled

-        When the samples were collected before or after mediation

Results

-        Please include the study limitations

-        Please add some figures of PCR of ETEC to the supplementary

-        The authors did not detect ETEC in the water source or perform CFs on the water to conclude strong associated between ETCE infection and water source

Discussion

-More discussion need to be added concerning the higher occurrence of LT.

- Add more intervention concerning the incidence of ETEC in HIV infected children

- Suggestions of educations programmers to teach prober hand washing could be beneficial in handling the ETEC diarrhea.

Comments on the Quality of English Language

Dear editor

Concerning the MS The incidence and risk factors for Enterotoxigenic E. coli diarrhoeal disease in children under 3 years old in Lusaka, Zambia

It includes the epidemiological data of Enterotoxigenic E. coli diarrhoeal disease in children under 3 years old in Lusaka, Zambia with some genetic data and serological studies of ETEC

Some points need to be elucidated

Introductions

More details are need to be added to the introduction concerning the toxicity of the EXEC, the types of toxins the virulence factors

Methods

-        Please indicate the inclusion and the exclusion criteria

-        How the colony PCR was performed

-        Mention the method, primers and used Kits

-        Is the children take any medication the study did not mention how the diarrhea was controlled

-        When the samples were collected before or after mediation

Results

-        Please include the study limitations

-        Please add some figures of PCR of ETEC to the supplementary

-        The authors did not detect ETEC in the water source or perform CFs on the water to conclude strong associated between ETCE infection and water source

Discussion

-More discussion need to be added concerning the higher occurrence of LT.

- Add more intervention concerning the incidence of ETEC in HIV infected children

- Suggestions of educations programmers to teach prober hand washing could be beneficial in handling the ETEC diarrhea.

Reviewer 3 Report

Comments and Suggestions for Authors

Manuscript ID: microorganisms-2857525

Title: The incidence and risk factors for Enterotoxigenic E. coli diarrhoeal disease in children under 3 years old in Lusaka, Zambia

The manuscript needs some modifications so that it could be better than before. It would be helpful if the authors would consider the following points:

Please highlight the advance of the study in Introduction. Please explain the development and creative work. The literature review should be carefully considered.

The hypothesis of the study should be clarified at the end of the Introduction section.

The various parts of the article are well articulated but the discussion needs to be improved by adding bibliography to confirm the results obtained. Argue the discussion well, comparing the results obtained with articles reporting similar work by showing the performance obtained in the other papers.

Enrich the discussion by addressing the study's limitations and practical applications.

Please polish the abstract conclusion section.

More related studies updated (2020-2024) should be considered when presenting literature review.

In title: Change " 3 years" to "three years"

In all pages: Change " Microorganisms 2023" to " Microorganisms 2024 "

Line 24: Delete "(cPCR)". This abbreviation was not written again in the abstract

Line 74: Change " 2.0km" to " 2.0 km"

Line 178: Change "have similar incidence" to "have similar incidence"

Line 190: Add "the" before "risk"

Line 298: Change "Piped" to "piped"

Line 312: "brining" ?. do you mean " bringing "?

Line 323: " Thea et. al " ?

Line 333: Change " highlights " to " highlighted"

In Table 1: Why are males " 5430 (79.5) " higher females " 1398 (20.5)" in "Head of Household sex"?

In Table 2: Please check " 2.45 (1.48-4.06) " as Multivariable aHR

In Table 5: Delete the empty cells

Lines 250-301: This paragraph is without references. Add a reference to support this paragraph

Insert the correct format style for journals in the references in the text and references list.

Reviewer 4 Report

Comments and Suggestions for Authors

The research conducted by Sukwa et al. aimed to establish the incidence of "ETEC" in children under 3 years of age in Zambia and identify possible risk factors for the occurrence of this microorganism through a prospective cohort study. The geographical area studied facilitates investigations into this pathogen, which holds global significance.

The topic is crucial, considering the substantial impact of this pathogen on public health in low- and middle-income countries (LMICs) and its worldwide relevance. The study is well-executed, moderately complex, and contributes to a better understanding of the epidemiology of this type of Escherichia coli.

The abstract section is well-written and comprehensive, presenting the authors' key findings.

The keywords need replacement for effective manuscript indexing, and authors are advised to avoid words already present in the title, opting for terms that better contextualize the study's field.

The introduction is relevant and concise, providing a comprehensive overview of the study's problem.

In the methodology, the inclusion of a figure illustrating the study area is recommended to aid international readers' visualization. Additionally, removing section 2.4 and distributing its content into appropriate topics (risk factors, questionnaire application, etc.) is suggested. In articles, a separate section solely for definitions is unnecessary.

The results section requires improvements in table presentation, while the discussion, although brief, is appropriate. The conclusion is supported by the obtained results.

Minor Remarks:

1.      Standardize the use of numbers in the manuscript according to the journal's guidelines. For example, Line 22 – 6828; Line 24: 6,828. Authors should follow the scientific journal's model.

2.      Remove the paragraph between lines 61 to 63. Conclude the introduction with the study's objective. The requested information removal will be presented throughout the manuscript (M&M and results – risk factors). Content between lines 63 to 65 can be integrated into the paragraph ending at line 57.

3.      Adjust the writing of ETEC in Figure 1.

4.      In Figure 1, ensure it is mentioned in the text before its appearance.

5.      Tables 1 and 2 need to be reformulated and resized for proper integration into the manuscript. Revise the presentation of these results.

6.      Line 321 – "the Other studies" must be referenced.

Reviewer 5 Report

Comments and Suggestions for Authors

In this study, authors have investigated on the incidence and risk factors associated with the Enterotoxigenic E. coli (ETEC) diarrhoeal disease among children below three years in Lusaka, Zambia. Overall, this study is worthy of investigation from the publica health implication perspectives and could be useful to the public health regulators to develop strategies to reduce the occurrence of ETEC-associated illness in young children. However, the reviewer has provided several comments for improvement of the overall quality of the manuscript.

Comments:

1)      The introduction is too brief. In the first paragraph of the introduction, could you provide some more background information about the public health implications of pathogenic E. coli (any current report on water-borne ETEC outbreak specifically in low- and middle-income countries?), also add a brief discussion on the pathogenicity characteristics/virulence attributes of ETEC.   

2)      Why you specifically interested on investigation of ETEC because there are also pathogenic E. coli (e.g., Shiga toxin–producing Escherichia coli) which can cause diarrhea.

3)      In the last paragraph of the introduction, please state what are the key hypothesis that were tested in this work.

4)      Line 116: “2.3. Laboratory procedures”. Provide the key information related to the colony polymerase chain reaction (PCR), the key technique used for confirmation of ETEC.

5)      Line 129 – 146: Add some reference to support the validity of the protocol used for the isolation and confirmation of ETEC.

6)      In the Materials and Methods, authors need to clearly state what control measures were taken to avoid the potential incidence of false positive as well as false negative results during ETEC isolation and confirmation steps.

7)      To improve readers understanding, authors are suggested to explain the abbreviations used in the tables in the footnote.

8)      At the end of discussion, add a paragraph on describing the limitations of this work.

9)      Please delete all the pronouns like we, our, us, etc. throughout the manuscript and change the text appropriately.  

Comments on the Quality of English Language

This manuscript could be benefitted with the minor English language editing.  

Round 2

Reviewer 5 Report

Comments and Suggestions for Authors

The submitted response to most of the comments are fine. 

Comments on the Quality of English Language

The English language of this manuscript looks good.